# Effect of Seasonal Variation during Annual Cyclist Training on Somatic Function, White Blood Cells Composition, Immunological System, Selected Hormones and Their Interaction with Irisin

**DOI:** 10.3390/jcm10153299

**Published:** 2021-07-26

**Authors:** Natalia Grzebisz-Zatońska, Stanisław Poprzęcki, Ilona Pokora, Kazimierz Mikołajec, Tomasz Kamiński

**Affiliations:** 1Warsaw College of Engineering and Health, Faculty of Cosmetology, Bitwy Warszawskiej 1920 Street 18, 02-366 Warsaw, Poland; 2The Jerzy Kukuczka Academy of Physical Education in Katowice, Institute of Sport Sciences, Mikołowska Street 72a, 40-065 Katowice, Poland; s.poprzecki@awf.katowice.pl (S.P.); i.pokora@awf.katowice.pl (I.P.); k.mikolajec@awf.katowice.pl (K.M.); t.kaminski@awf.katowice.pl (T.K.)

**Keywords:** cycling, training, immunology, hormones, WBC, somatic structure, irisin

## Abstract

The aim of this study was to evaluate somatic, hormonal and immunological changes during the macrocycle of cyclists (9 well-trained men, age 25.6 ± 5.2 years and body weight 72.4 ± 7.35 kg). During the training macrocycle, four exercise control tests were carried out, and biochemical markers were measured in the laboratory. Seasonal training changes did not significantly disturb resting somatic and functional parameters, physical capacity (VO2max), body weight, the number of leukocytes and selected hormones. The secretory system of the organism did not respond significantly to the exercise stress in the training process, even with the increasing share of anaerobic processes in the subsequent periods of the macrocycle. Irisin and other parameters globally did not correlate with training volume. Irisin showed a significant correlation only with cortisol in the first period and human growth hormone in the second, and it showed a weak correlation in the third period with body mass and BMI. The lack of interactions between irisin level and other variables practically excludes its use in monitoring cyclist training. Future research would be complemented by the assessment of stress and postexercise changes in the cyclists’ macrocycle and expanding the research group to other athletes, including women.

## 1. Introduction

Physical training is a long-term process aimed at producing adaptive changes in many body systems, including the circulatory, respiratory, endocrine and immune systems, as well as the adjustment of somatic features, which, together with systemic and metabolic changes, will contribute to the improvement of the athlete’s sports performance during competition [1]. The stimulus for achieving these changes is a sufficiently strong exercise stress, regulated by the volume of training and its intensity [2], as well as its temporal structure [3,4,5]. Exercise metabolism can be modified by environmental conditions, the level of energy substrates that affect the control of hormonal balance by increasing the level of, e.g., testosterone (Test), cortisol (Cort), estradiol, human growth hormone (GH), insulin-like growth factor-1 (IGF-1) and irisin (IR) [6,7]. In addition, they are able to modify the activity of the immune system, mainly through the anti-inflammatory response mediated by cytokines [8,9], the composition and volume of circulating blood, as well as the creatine kinase (CK) and lactate dehydrogenase activity (LDH) activity of muscle damage markers [10,11,12]. Training can differentiate the effectiveness of induced adaptive changes in various systems, also taking into account oxidative stress [13,14].

Skeletal muscles, apart from locomotor functions, have secretory abilities, e.g., secretion of cytokines [15] and hormones [16,17]. One of the secretions of the muscle cell in response to exercise is IR, discovered and described by [18] and classified as a myokine. IR alleviates the effects of oxidative stress and controls the energy homeostasis of myocytes and adipocytes, inducing thermogenesis and the transformation of white into brown (beige) adipose tissue [19,20]. It also increases energy expenditure by stimulating the expression of the uncoupling protein 1 (UCP1), mainly in brown (beige) adipocytes [18,21]. This hormone reduces the level of tumor necrosis factor alpha (TNF-α) and other inflammatory factors [22,23]. Recent studies have shown that IR is also an adipokine released from adipose tissue [24,25].

Long-term exercise can lower circulating IR levels, although [18] observed a 2-fold increase in plasma IR concentration in eight men after 10 weeks of endurance training at 65% maximum oxygen uptake (VO2max). Other studies have not confirmed a significant impact of endurance exercise on the blood IR level [16,26,27]. Inconclusive and often contradictory results regarding the effect of training on irisin production have been reported [18,23,28]. On the other hand, single high-intensity efforts increased the concentration of IR in the blood [27,28,29]. The likely reason for the increase was a significant share of eccentric muscle contractions during these forms of exercise [23,30,31]. The quality of training affects skeletal muscles by activating, among other things, their secretory abilities, e.g., cytokines and other hormone-like proteins. This can increase the effectiveness of communication between muscle fibers and organs and promote training effects [8,15]. In addition, other studies have shown that IR is positively correlated with body mass index (BMI), lean mass index (FFMI) [30,32] and IGF-1 and negatively correlated with insulin and thyroglobulin levels, and according to researchers, it may participate in compensatory metabolic regulation [28,33]. In [28], it was shown that the increase in blood IR levels was accompanied by an increase in the activity of the CK enzyme, a marker of muscle damage. The influence of exercise on the production of IR is far from being fully understood, despite the classification of IR into the group of exercise hormones. Considering the above, the changes in blood IR concentration may be influenced by the quality and temporal structure of the training loads used in cycling. In addition, the training process can differentiate many somatic, biochemical, energetic, hematological, hormonal or immunological variables. These changes may be related to the exercise characteristics of the training period and changes in resting IR concentration.

The aim of the study was to evaluate somatic and functional characteristics, CK and LDH activity, concentration of malondialdehyde (MDA), lactate level (LA), levels of selected hormones, changes in the differential blood count and the level of immunological parameters in four periods of the training macrocycle of road cyclists and to demonstrate their relationship with the concentration of irisin in serum. Moreover, the aim was to demonstrate the dependence of these variables on the training load of cyclists at individual stages of the training macrocycle.

## 2. Materials and Methods

The study involved 9 well-trained cyclists aged 25.6 ± 5.2 years, weight 72.41 ± 7.35 kg, body height 181.44 ± 5.64 cm, body mass index (BMI) 21.94 ± 1.49 kg/m^2^, fat content 10.73 ± 2.78%, and maximum oxygen uptake (VO2max) 65.78 ± 3.87 mL/kg/min. The inclusion criteria were physical capacity in the order of 5 L/min, professional participation in sports and training experience of 5–7 years. The week before each time trial test, the cyclists did not change their eating habits (isocaloric diet) and did not consume supplements. The day before the test, they did not supplement their diet with caffeine and did not perform intense efforts 48 h before the test. The last hydration of the body took place 60 min before the exercise test. The training process was based on the concept of linear periodization and included 4 critical training stages from November to July of the following year. Test T0 (TRANS) took place after the transition period (November/December), test T1 (PREP) took place in the preparatory period (January/February), test T2 (COMP I) took place at the beginning of the competition period (April) and test T3 (COMP II) took place during the competition period (June/July). In each of them, the anthropometric characteristics and the level of aerobic capacity (VO2max) of the competitors were assessed. In addition, blood was collected each time at rest for biochemical analyses. The research project was carried out in accordance with the Helsinki Declaration, and its course was approved by the Scientific Research Ethics Committee, Jerzy Kukuczka Academy of Physical Education in Katowice, Poland (No. 2/2014).

Training loads at individual stages of the research were individually recorded by the competitors using a heart rate monitor (Garmin, Olathe, KS, USA). They were calculated after each training session and archived using the WKO + 4.0 software (Training Peaks, Louisville, TP, USA). Table 1 shows the average training load in each period of the training macrocycle, taking into account the percentage of work in the zone of aerobic, mixed and anaerobic metabolic changes, while Figure 1 shows the sums of monthly training loads between the stages of tests T0–T1, T1–T2 and T2–T3. Figure 2 shows average share of aerobic, mixed and anaerobic energy sources during the macrocycle of cyclists.

During the training macrocycle, four exercise control tests were carried out in the laboratory: T0 (baseline) after the transition period (TRAN), T1 after the beginning of the preparation period (PREP), T2 at the beginning of the competition period (COMP I) and T3 during the competition period (COMP II). Before each test, the body height (in cm) was assessed with an anthropometer with an accuracy of 0.5 cm. Body weight (in kg) was assessed with an accuracy of 100 g. Body composition was determined by the electric impedance method using the In Body 570 analyzer (InBody Co., Ltd., Seoul, Korea). The body mass index of the cyclists was calculated using the parameters of body weight and height^2^ (BMI kg/m^2^). During the macrocycle, the competitors used a fixed diet tailored to their individual needs, the composition of which was not changed before the stress test, and an individually programmed training program that was recorded by the cyclists using a Garmin heart rate monitor. The authors did not participate in the planning and corrections of the training tasks.

The VO2max control test was performed on the same LODE Excalibur Sport bicycle ergometer (Lode BV, Groningen, The Netherlands) which was coupled to a MetaLyzer 3B-R2 gas analyzer (CORTEX Biophysik GmbH, Leipzig, Germany). The test was carried out in laboratory environmental conditions, temperature 19–21 °C and relative humidity 40–50%, and at the same time of the day. The standard VO2max test was performed with an initial load of 40 W, after which the load was gradually increased by 40 W over a period of 3 min until refusal, i.e., until the turning cadence was not maintained at 70–75 rpm (rpm). During the exercise test, the following were recorded: oxygen uptake and VO2max.

### 2.1. Analytical Methods/Biochemical Determinations in Blood at Rest

Athletes began the exercise test on an empty stomach, each time under standard laboratory conditions. Blood for biochemical determinations was collected 30 min before the test, in the morning, from the antecubital vein in a sitting position. Each time, 10 mL of blood was collected: 8 mL in a test tube without anticoagulant and 2 mL with EDTA, using the Vacutainer technique. Serum was obtained by centrifugation at 2000× *g* and 4 °C for 15 min, and serum samples were subsequently stored at −20 °C for future biochemical analysis.

### 2.2. Whole Blood Hematology Determinations

The number of white blood cells (WBCs) per volume unit (µL) was determined with a Cell Dyn Ruby apparatus (2015), Abbott Laboratories; the results are given in 10^3^/µL. The analysis included WBCs, lymphocytes (Lymph), monocytes (Mono), neutrophils (Neut), basophils (Baso) and eosinophils (Eos). Lactate concentration (mmol/L) was determined in capillary blood collected from a fingertip using a device (Biosen C-Line Clinic, EKF-Diagnostic, Cardiff, UK).

### 2.3. Immunological Parameters

Immunoglobulin A (IgA, g/L) was determined with the immunoturbidimetric method and the diagnostic kit IGA 2073755 322 using the Cobas Integra 400 plus biochemical analyzer (Roche Diagnostics, Basel, Switzerland), and immunoglobulin G (IgG, g/L) was determined with the IGGT kit (turbidimetric method). C-reactive protein (CRP, mg/dL) was determined by immunoturbidimetric method with the reinforcement of latex particles, using the diagnostic kit C-Reactive Protein Latex (CRPLX) and the Cobas Integra 400 plus biochemical analyzer; the precipitate was measured at the turbidimetric wavelength of 552 nm (Roche Diagnostics, Basel, Switzerland). The concentration of interleukin-6 (IL-6, pg/mL) was measured with the Human IL-6 High Sensitivity ELISA KIT Cat diagnostic kit (No. 950.035.096, Diaclone SAS, Besancon, France). Tumor Necrosis Factor α (TNF-α, pg/mL) was measured with the Human TNF α ELISA KIT (Catalog No. 1x90 950.090.096, Diaclone SAS, Besancon, France).

### 2.4. Hormone Concentration

GH was assessed in the serum using the Beckman Coulter IV D IRMA GH Ref. IM 1397 kit (µg/L), using the immunoradiometric method (ImmunoTech s.r.o., Praha, Czech Republic). IR concentration (0.2–2 µg/mL) was determined by the ELISA system (kit) BioVentor- Laboratorium medicina as Irisin ELISA (Catalog No. RAG018R, Czech Republic). IGF-1α (ng/mL) was measured with the Beckman Coulter IV D IRMA IGF-1 kit Ref. A15729 (ImmunoTech s.r.o., Praha, Czech Republic), Cort (µg/dL) with the CORTISOL (125I) RIA KIT (Ref: RK-240CT). Test (Total T ng/mL) was determined with the reagent kit, Testosterone (I-125) RIA KIT (Ref: RK-61CT). The Testosterone (125I) radioimmunoassay system provides the quantitative in vitro determination of testosterone in human serum (range 0–60 nmol/L), using 25 µL serum sample (Institute Of Isotopes Ltd., Budapest, Hungary). The testosterone/cortisol (T/C) ratio was then calculated. Moreover, serum CK activity (EC2.3.7.2., IU/L) was determined with the RANDOX diagnostic kit (Randox Laboratories Ltd., Crumlin, UK). The concentration of malondialdehyde µmol/L) was determined according to the method described in [34].

### 2.5. Statistical Analysis

The results were expressed as mean values (X) ± SE or standard deviation (SD) and percentage changes in the resting values of the tested variables in individual training periods (T1–T3) relative to T0, according to the formula %∆RV = ((XT1−T3−XT0)/XT0 × 100). The Kolmogorov–Smirnov test and Levene’s homogeneity of variance analysis were performed to verify the normality of the distribution of variable values. As a statistical evaluation tool, one-way analysis of variance (ANOVA) was employed after checking the homogeneity of variance with Levene’s test; the main factor “test”, i.e., training period/stage in the training macrocycle (T0, T1, T2, T3); and the strength of the η^2^p effect (from 0 to 1). In the post hoc analysis (when F values indicated *p* < 0.05), the Bonferroni test was used to establish between which periods (stages) of training significant differences in the values of the dependent variables were noted. Pearson’s linear correlation test (r) was used to establish the relationship between the concentration of irisin and the studied variables. Moreover, the Pearson linear correlation test (r) was used to assess the relationship between the training load and the variables included in the study. The statistical analysis of the variables was performed on a PC using the statistical package Statistica v. 10 Pl. The significance level of differences for all analyses was set at *p* < 0.05.

## 3. Results

### 3.1. Somatic and Functional Parameters

The training process did not significantly affect the value of body weight (BM) (F = 0.213) or BMI (F = 0.396). However, there was a tendency towards a decrease in the values of these features in the T3 period (Table 2, Figure 3). 

The analysis of variance showed a significant effect of the training period (“test”) on the fat content (%) (FAT%) (F = 3.297, *p* = 0.037, η^2^p = 0.292). In the T3 period, there was a significantly lower FAT% (by 12.44%) (*p* < 0.05) compared to T0. A similar tendency was found in the FAT (kg) parameter (Table 2, Figure 3). The training process did not significantly differentiate VO2max (F = 2.518). During the training macrocycle, the VO2max level fluctuated, reaching a value 6.18% higher in the starting period than in the T0 period (Table 2, Figure 3). The training period did not significantly affect the concentration of MDA (F = 0.448), although in T2 and T3 the level of this metabolite was 19.90% higher than in T0 (Table 2, Figure 3). The training significantly influenced the activity of CK (F = 3.662, *p* = 0.026, η^2^p = 0.314). In T3, CK activity was significantly lower than T2 (*p* < 0.05), and compared to T0, it was lower by 10.27% (Table 2, Figure 3). The training program did not significantly affect the activity of LDH (F = 1.817). In the T3 period, LDH activity was 6.22% higher compared to T0. The training period did not significantly differentiate LA concentration in the blood (F = 2.379). In the training macrocycle, a successive decrease in LA concentration was observed, which in T3 was 14.85% lower than in T0 (Table 2, Figure 3). The independent variable “test” did not significantly affect the concentration of irisin at rest (F = 2.737), but in T3 its level was 21.37% higher than in T0 (Table 3, Figure 4).

During the macrocycle, no significant correlations between IR and somatic–functional parameters were found, and the strength and direction of this relationship were varied. In the starting period (T3); insignificant but high positive correlations were noted between the concentration of IR and BM (r = 0.612, *p* = 0.08) and between the concentration of IR and BMI (*p* = 0.09). There was no significant correlation between the IR concentration and VO2max (Table 4).

There were also no significant correlations between IR and MDA (r = 0.168), CK (r = 0.523), LDH (r = 0.372), and LA (r = 0.156) (Table 4).

### 3.2. Leukocytes and Leukocyte Subsets

The training did not significantly affect the number of WBCs (F = 2.246). However, in the competition period, the number of WBCs was 7.34% higher than in T0. The independent variable “test” significantly differentiated the counts of Lymph (F = 3.961, *p* = 0.020, η^2^p = 0.331), Mono (F = 6.689, *p* = 0.002, η^2^p = 0.445) and Baso (F = 12.306, *p* = 0.000, η^2^p = 0.606). In T3, the Lymph count was 33.57% higher than the T0 value (*p* < 0.05), Mono was 55.28% higher than the T0 value (*p* < 0.05) and Baso was 77.60% higher than the T0 value (*p* < 0.001). Similarly, training significantly influenced the Baso number (F = 12.306, *p* = 0.000, η^2^p = 0.606). In T3, the Baso number increased by 77.60% in relation to the T0 value (*p* < 0.001). The training period did not significantly differentiate the number of Neut (F = 1.841) or Eos (F = 0.478). However, in T3 the numbers of Neut and Eos were 26.41% and 7.99% higher, respectively, than in T0 (Table 3, Figure 5). The numbers of WBCs and subpopulations of white blood cells were not significantly correlated with the concentration of IR at rest (Table 4); only a weak inversely proportional relationship between the concentration of IR and the number of Eos (*p* = 0.058) and Baso (*p* = 0.06) in the period T1 and T2 was noted.

### 3.3. Immune System

The training program significantly differentiated the CRP concentration (F = 14.32, *p* = 0.000, η^2^p = 0.641). At T3, the CRP concentration was 34.36% higher than that of T0 (*p* = 0.000). The training process did not significantly affect the concentration of cytokines TNF-α (F = 2.808) and IL-6 (F = 0.527). However, in T3, the concentration of TNF-α was 15.26% lower than in the T0 period, while IL-6 was 7.22% higher in relation to T0. Training did not significantly affect the levels of IgG (F = 1.135) and IgA (F = 0.493) (Table 3, Figure 6). There was no significant correlation between IR and the studied immunological factors (Table 4).

### 3.4. Hormones

The cyclists’ training program had a statistically significant effect on the concentration of cortisol (F = 5.308, *p* = 0.006, η^2^p = 0.399). In the T3 period, its level was 16.89% (*p* < 0.05) lower compared to T2 and 8.66% lower compared to T0. Training did not significantly differentiate the concentrations of testosterone (F = 0.252), GH (F = 0.481) or IGF-1 (F = 2.267), nor did it significantly differentiate T/C (F = 1.046). In the starting period (T3), however, it was noted that the concentrations of IGF-1, GH and testosterone were 17.11%, 25.37% and 8.64% higher than T0 levels, respectively, and the T/C ratio was 19.18% higher than that at T0 (Table 4).

A significant positive correlation was found between the IR concentration and the cortisol concentration in the T0 period (r = 0.758), and a negative correlation was found between the IR concentration and the GH concentration in T1 (r = −0.759). The remaining hormones were not correlated with the IR concentration (Table 4).

### 3.5. The Strength of the Relationship between Training Load and the Studied Variables

A significant correlation between training load and selected variables in individual stages of the study was demonstrated: testosterone in T2–T3 (r = 0.907, *p* = 0.001); T/C in T2–T3 (r = 0.938, *p* = 0.005); Eos in T2–T3, negative relationship (r = −0.771, *p* = 0.015); VO2max in T2–T3, negative relationship (r = −0.798, *p* = 0.010); and TNF-α in T2–T3 (r = 0.682, *p* = 0.045). On the other hand, an insignificant moderate relationship between the training load and the analyzed variables occurred among the following variables: Cort in T0–T1 and negative in T1–T2, GH in T2–T3, CRP in T1–T2, Eos negative in T1–T2 and Neu in T1–T2.

## 4. Discussion

### 4.1. Somatic, Functional and Biochemical Changes during the Training Macrocycle

In this study, the training process of cyclists during the macrocycle significantly differentiated the percentage of body fat (*p* < 0.05). Overall (globally) the fat content during the macrocycle successively decreased in relation to T0, and the percentage decrease in FAT% in T3 vs. T0 was statistically significant (*p* < 0.05). However, training did not significantly affect BM or BMI. During the competition period (T3), the value of these parameters slightly decreased compared to the transition period (T0). The decrease in muscle mass, including adipose tissue, in the competition season could be the cause of a reduction in their secretory activity. According to [35], a decrease in the body fat content measured by the sum of the thickness of skin folds with a simultaneous increase in body weight is associated with an increase in physical capacity. Similar trends were observed in [10], examining 12 cyclists in the annual training cycle, i.e., from November to June of the following year. The body weight of the cyclists practically did not change during the preparations for the competition and amounted to approx. 70 kg, while the percentage of fat content in the preparation and competition periods decreased significantly compared to the transition period (*p* < 0.05). Moreover, in our study, the training process significantly differentiated the CK activity (*p* = 0.026); a significant decrease in the activity was found in the T3 study compared to the T2 study (*p* < 0.05). It can be stated that with the development of athletes’ adaptation to effort during training, microdamage to skeletal muscles is reduced and oxidative stress is reduced, which is often associated with the improvement of the efficiency of the defense system against reactive oxygen species and the effects of inflammatory factors [36]. As a consequence, the tight structure of the cell membrane is maintained, which is accompanied by a lower release (leakage) of CK and LDH cellular enzymes to the environment. In this experiment, the CK activity of cyclists was low in relation to athletes, i.e., on the order of 140 U/L, with the norm of 24–195 U/L for men. However, the training process did not significantly differentiate the activity of resting LDH during the macrocycle of cyclists, and its course of changes was similar to CK. It is known that the body of well-trained athletes is less responsive to training load, which is manifested by stable serum CK and LDH activity, compared to individuals not undertaking physical activity [37,38]. The authors of [39] investigated the resting activity of CK three times during the speed skater preparatory season in June, September and November. In the second preparatory period (September), the activity increased significantly, which was probably a response to the high training load. Then, in the precompetition period, adaptation to exercise was observed, manifested by a decrease in CK activity to a value slightly below the baseline. An interesting phenomenon is the resting changes in lactate levels during the cyclists’ macrocycle. In the present study, the variable “training period” did not significantly differentiate the resting LA level. The highest values were obtained in the T0 test, and then the LA concentration gradually decreased despite the greater involvement in the training of anaerobic metabolism and greater load at this stage of the training. This may have positive consequences for the fatigue process, as LA and the concentration of protons regulate the exercise physiological adaptations of the system [40].

In general, cyclists are characterized by high physical performance measured by VO2max. In this experiment, cycling training did not significantly affect the level of aerobic capacity assessed by the value of the VO2max, although during the training process there was a tendency towards an increase in efficiency in the T3 vs. T0 test. This is confirmed by the authors of [10] and by other authors who showed that well-trained road cyclists with a high, 60–70%, VO2max were less responsive to the training stimulus aimed at increasing VO2max and their physical performance practically did not change during the macrocycle. In the present study, the resting MDA value was not significantly differentiated during the macrocycle; however, an insignificant trend towards its increase in the starting period was demonstrated (COMP II, T3). Similar results were obtained by the authors of [13] in well-trained endurance cyclists who were not exposed to oxidative stress during training sessions, also in conditions with and without supplementing the diet with antioxidant vitamins. On the other hand, in [14,41], the presence of oxidative stress was observed during high-intensity exercise (VO2max test) in training cyclists, while no such tendency was shown during submaximal exercise. Together, these findings emphasize the importance of the basic training of athletes in the occurrence of this phenomenon. In addition, oxidative stress factors regulate exercise physiological adaptations in the body [42]. In these studies, the somatic parameters, activity of indicator enzymes and MDA concentration did not show a significant relationship with the training load at individual stages of the study, except for the starting period (T2–T3), which showed a significant negative relationship for VO2max (r = −0.875, *p* = 0.010).

### 4.2. The White Blood Cell System and Selected Indicators of the Immune System in the Cyclists’ Training Macrocycle

It is known that the number of WBCs in the blood increases after a single exercise, depending on the intensity, duration and type of exercise. In addition, the number of WBCs in the blood is modified by age, training status, diet and ongoing infection [43,44]. In this study, the resting numbers of WBCs and their components during the cyclist macrocycle were not stable. The training significantly differentiated the numbers of Lymph (*p* = 0.020) and Mono (*p* = 0.002). During the T1 and T3 tests, a nonsignificant decrease in the number of white cells was noted. In the T1 study, the number of Mono decreased significantly compared to T0 (*p* < 0.05). In the T3 test, an insignificant increase in Neut and Baso and a significant increase in the number of Lymph were found (*p* < 0.05). In general, trends in the numbers of WBCs and their components were similar to those observed in the activity of CK in response to multiweek cycling training. Long-term follow-up of WBC blood count by Horn et al. [45] showed adaptive changes in the white blood cell image in various groups of athletes consisting in a reduction in the number of WBCs in aerobic sports, including cyclists. The content of WBCs and Neut in about 16% was lower than the physiological norm. Overall, mechanical stress had a lesser effect on the distribution of these cells. In addition, the number of WBCs circulating in the vessels is influenced by exposure to external pathogens, oxidative stress and the presence of inflammatory mediators [46,47].

In [48], during a 5-day field trial of professional cyclists, after covering an average distance of 750 km, an increase was shown in the number of leukocytes. On the other hand, [47] revealed an insignificant decrease in the number of WBCs by 7.28% during the volleyball season. Exercise during the triathlon caused a significant increase in the number of leukocytes (*p* < 0.05), which was maintained during the restitution period in both the trained and amateur groups. A similar phenomenon was observed in the case of neutrophils, lymphocytes and monocytes [49]. In [13], no significant resting changes were found in the numbers of WBCs, Lymph, Mono, Neut, Eos and Baso in well-trained cyclists during the preparation period and in the initial and final stages of the racing period, i.e., in the 10th, 19th and 29th weeks of the training cycle. In this study, no significant relationship was found between the training load and the white blood cell system, except for Eos, where a negative statistically significant relationship was observed (r = −0.771, *p* = 0.015) in the starting period (T2–T3).

In the present study, training in the cyclists’ macrocycle did not significantly differentiate selected immunological parameters—TNF-α, IL-6, CRP, IgA and IgG. However, there was a tendency towards a nonsignificant increase in CRP concentration, while the levels of IgA and IgG decreased by 6.27% and 7.83%, respectively, and the concentrations of TNF-α and IL-6 decrease by 16.30% and 8.45%, respectively, in the starting period (T3) compared to the baseline value (T0). There is a difference of opinion in the literature on the impact of the quality of the training stimulus on the level of the components of the immune system. In [50], an insignificant increase was shown in the concentrations of IL-6, IL-1β and TNF-α after 6 weeks of resistance training, while the level of IL-6 decreased significantly after endurance training in overweight men. Similarly, in other studies during resistance, endurance and mixed training in nontraining subjects, no significant changes in the concentrations of inflammatory factors IL-6, TNF-α and CRP were found [51]. IL-6 belongs to the multifunctional cytokines that play an important role in the regulation of immunity, and it is a cytokine with two functions; i.e., it has proinflammatory and anti-inflammatory effects. During a single exercise, the level of IL-6 increases exponentially depending on the intensity, duration and recruitment of muscle mass [52,53,54]. Working skeletal muscle appears to be one of the sources of IL-6 in the circulation [55] in addition to adipocytes, macrophages, monocytes, lymphocytes, connective tissue cells and others and mediates communication between immune cells, organs and the organ system during a training session, which may indicate the stability of the immune system as the “sports form” increases [56]. Another protein in this system is CRP, which shows high variability in blood concentration during training sessions, and a tendency to increase in concentration was observed after single efforts [57]. Additionally, [13] reported low serum IgA levels during the cyclists’ preparatory season, explaining this phenomenon with chronic immunosuppression in response to training loads, work volume and intensity and increasing adaptation to exercise. A similar tendency was observed by Gleeson [44,58] in swimmers during 1.5 h of moderate- to high-intensity exercise. The authors of [47] studied volleyball players for 4 months of the starting season; the concentration of IgG in rest at the end of the season decreased by 18%, and that of IgA decreased by 5%. On the other hand, a single exercise test significantly increased the level of IgG before and after the season, and the concentration of IgA was insignificantly increased. The increase in the level of IgG during rest in volleyball players was a consequence of (post)stress inflammatory processes stimulating an increase in the number of circulating lymphocytes and cortisol concentration both after a single exercise and after the accumulation of exercise stress in the competition season. Taking into account the data of other authors, it can be concluded that a long intense effort weakens (suppresses) the immune system, while a short one of medium intensity strengthens its functions. In this study, a significant dependence of TNF-α level was found on the training volume in the starting period (T2–T3). The other immunological variables did not show any significant dependence on the training volume.

### 4.3. The Concentration of Selected Hormones in the Training Macrocycle

The quality of training affects skeletal muscles by activating, among other things, their secretory abilities, e.g., cytokines and other hormone-like proteins. This may increase the efficiency of communication between muscle fibers and organs and promote training effects [15,59]. Decreases in testosterone levels were found after a long single effort at low temperature [60] and after a few days of training [61,62]. However, during prolonged exercise, the levels of GH and cortisol increased significantly [60,63]. In the present study, the resting hormone levels were stable during the cyclists’ training macrocycle. Only training in the T0–T1 period had a significant effect on the level of cortisol (*p* = 0.006). In T3, the level of cortisol significantly decreased compared to the T2 test (*p* = 0.05), indicating a general reduction in the size of the stress response to training loads used in the preparation season. The decrease in cortisol secretion in the training process was accompanied by an increase in GH concentration. In general, during the competition period (T3), an insignificant increase in the resting levels of IGF-1 hormones, GH, testosterone, the T/C ratio and serum irisin was observed. This may indicate the stability of the sarcolemma structure caused by training, as evidenced by low resting CK activity and a stable level of MDA.

Generally, exercise stress reduces testosterone levels and increases cortisol levels, but to a different extent, so the T/C ratio was used as an indicator to assess the state of fatigue and overtraining in terms of the anabolic/catabolic state of the system [64], although the results of studies in this regard are not convincing [65]. Heavy resistance training is a signal for an increase in testosterone, GH, cortisol and adrenaline levels, while the size of these changes depends on the volume and duration of exercise [7,66,67]. In the present study, the concentration of selected hormones did not significantly depend on the training volume, except for testosterone concentration, which was significantly correlated with the training volume in the starting period (T2–T3, r = 0.907, *p* = 0.001).

### 4.4. Irisin Concentration in the Training Process (Macrocycle)

In the assessed stages of the training macrocycle of this experiment, no significant influence of the main factor “test” on the concentration of irisin in the blood serum was demonstrated. On the other hand, a higher tendency of its concentration was revealed in the precompetition and competition period compared to the baseline stage (T0). In [59], it was shown that the quality of training affects skeletal muscles, including triggering their secretory abilities, e.g., cytokines (myokines) and other hormone-like proteins. They are used to communicate with other fibers and organs, which favors beneficial training effects [15]. In [68], it was shown that a single exercise increased blood levels of irisin. This mainly concerned the resistance effort [28,29]. The increase was probably due to the involvement of fast-twitch fibers during eccentric effort [31]. However, according to [69], the induction of changes in blood IR concentration during endurance exercise is questionable. In [70], an increase in irisin production was shown during eccentric work with an intensity of 70% VO2max for 30 min. In addition, heavy muscle work can cause myokine proteolysis; therefore, excessive muscle damage during eccentric work may temporarily weaken IR production [71]. Globally, the literature shows that resistance effort increases the concentration of IR, while endurance effort does not cause changes in its content in the serum. Thus, it is difficult to talk about the exercise benefits attributed to IR associated with the endurance effort of cycling [72].

Moreover, the data of dependent variables concerning somatic, functional or biochemical traits recorded in the present study were differentiated by the training process, but in general, they were not strongly related to the IR concentration. It was only in the competition period (T3) that the IR concentration was correlated more strongly with BM and BMI. Nygaard et al. [29] showed that the secretion of irisin by muscle and fat cells depends on body weight, type of muscle work and exercise intensity. Moreover, in the present study, no significant correlation of IR with resting CK and LDH activity and LA level was found. On the other hand, the direction of dependence during the T0 test was negative, and in the starting period (T3) it was positive. The strength of the relationship between IR and CK in the starting period was high (r = 0.523), while with LDH it was low; moreover, a weak relationship between IR and LA was noted (r = 0.156). The dependence of the IR level on the concentration of MDA was statistically insignificant. These variables were negatively correlated in the T0 test and positively correlated in T3. In addition, during the analyzed stages of the cyclists’ macrocycle, no significant relationship between IR concentration and VO2max was found. At each stage, a low relationship with a negative direction was recorded. In this study, no significant correlations between irisin and WBCs and their components were found. All relationships in the T3 study (competition period) had a negative direction; similar results were found in the transition period (T0), except for the number of basophils. The strength of the relationship between irisin and white blood cell counts in the transition period (T0) was weak and average, while it was average in the baseline period (T3). Similarly, no significant correlation was found between the level of irisin and selected immunological parameters. During the transition period (T0), a weak correlation between IR and TNF-α, IL-6 and negative IgG was noted (r = 0.300), while there was no correlation in the case of IgA (r = −0.058). Moreover, in the T3 study, the correlation of negative variables concerned medium-strength TNF-α, similarly to IgA-positive drugs. There was no correlation between IR and CRP, IL-6 and IgG proteins in the competition period. However, serum IR concentration significantly was correlated with cortisol in T0 (r = 0.758) and GH in T1 (r = −0.759). The dependence of testosterone and cortisol on IR was positive in T0, and in the competition period (T3), testosterone was negative, and cortisol was positive. In the T0 test, IR was negatively correlated with T/C, GH and IGF-1; similar results were found the T3 test. This differentiation may indicate the relationship between IR and the increased catabolic processes at that time, the high metabolic load, the increased peroxidation of membrane lipids measured by changes in MDA concentration and the predominance of anaerobic processes in the characteristics of training loads used in the starting period (T3). Similarly, there was no significant relationship between IR and training load. The highest correlation was observed at the T0–T1 stage (r = 0.514). In the remaining stages of the training macrocycle, there was practically no relationship between IR and training load (r = 0.2 and r < 0.2).

### 4.5. Limitations

The methodological limitations of the test results were related to the small number of athletes who completed the experiment in the planned exercise tests and the monitoring of training loads by the athletes. In addition, the cyclists represented various sports clubs. During the competition period, their training loads were supplemented with starts in competitions, which in turn made it difficult to determine the actual loads precisely. Higher training loads in the competition period resulted in a stronger exercise stimulus than in the previous periods. Apart from that, the authors of the study were not involved in the planning and implementation of the training. Expanding the research group to other athletes, including women, with similar exercise capacity would also be a valuable enrichment of this research. Future research would also be complemented by the assessment of stress and postexercise changes in the cyclists’ macrocycle.

## 5. Conclusions

The aim of this study was to evaluate somatic, hormonal and immunological changes as a result of intentional exercise adaptation of the body during the macrocycle of road cyclists. The metabolic and hormonal response of the body to physical training showed that reducing the volume of training with a simultaneous increase in work intensity before the competition period was probably beneficial for exercise adaptation. Serum irisin concentration and other parameters globally did not correlate with training volume. Seasonal training changes generally did not significantly disturb resting somatic and functional parameters and physical capacity (VO2max). In the competition period, the body weight decreased insignificantly, and the physical capacity increased, while the numbers of leukocytes and their subsets (leukocyte subset count) increased. Nevertheless, the exercise stress accompanying the training significantly differentiated the fat content (%) and CK activity, as well as the numbers of lymphocytes and monocytes, and the level of TNF-α clearly increased. A resting increase in IgG levels may indicate an increase in the resistance of the respiratory system to pathogens accompanying increased lung ventilation during exercise. The secretory system of the organism did not respond significantly to the exercise stress used in the training process, even with the increasing share of anaerobic processes in the subsequent periods of the training macrocycle. Irisin practically showed a significant relationship (correlation) only with the levels of cortisol (T0) and GH (T1), but it was also weakly correlated with BM and BMI in T3. In general, the lack of significant biochemical and physiological consequences of irisin and the lack of relationships (interactions) between irisin level and other variables practically exclude it from involvement in long-term endurance cycling effort.

## Figures and Tables

**Figure 1 jcm-10-03299-f001:**
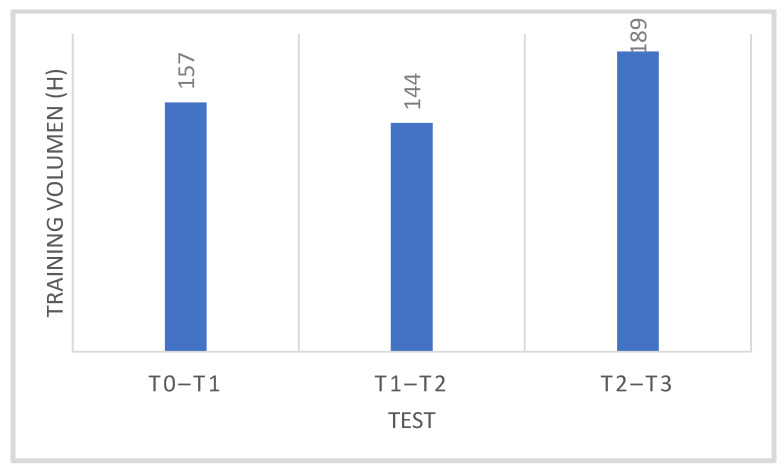
Summary of monthly training load (in hours) in each test period—T0–T1 (Nov, Dec, Jan), T1–T2 (Feb, March) and T2–T3 (Apr, May, Jun).

**Figure 2 jcm-10-03299-f002:**
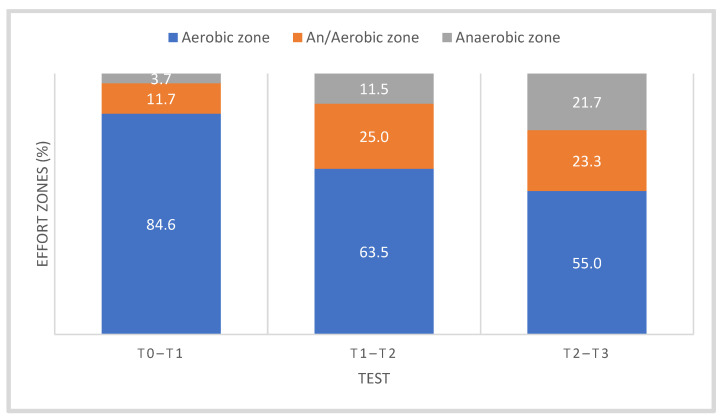
Average share of aerobic, mixed and anaerobic energy sources during the macrocycle of cyclists, including training stages—T0–T1 (Nov, Dec, Jan), T1–T2 (Feb, March) and T2–T3 (Apr, May, Jun). Abbreviations as in Table 1 and Figure 1.

**Figure 3 jcm-10-03299-f003:**
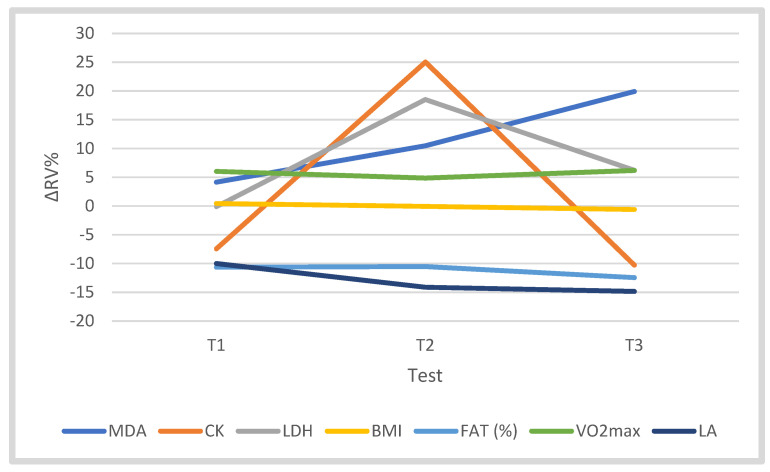
Changes (∆%) in somatic–functional parameters (BM, BMI, FAT%, VO2max); the concentration of MDA; and LA, CK and LDH activity of cyclists in tests T1, T2 and T3 in the training cycle, relative to T0; ∆RV% = ((XT1−T3−XT0)/XT0 × 100).

**Figure 4 jcm-10-03299-f004:**
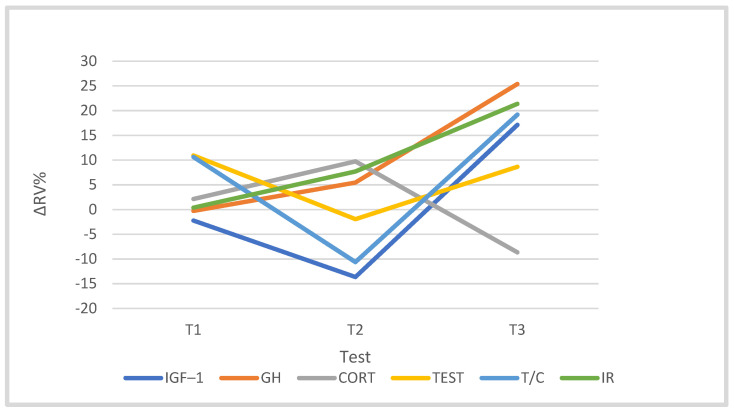
Changes (∆%) in hormonal indices in the cyclists’ blood in the training macrocycle in tests T1, T2 and T3, in relation to the baseline test (T0). Abbreviations as in Table 3.

**Figure 5 jcm-10-03299-f005:**
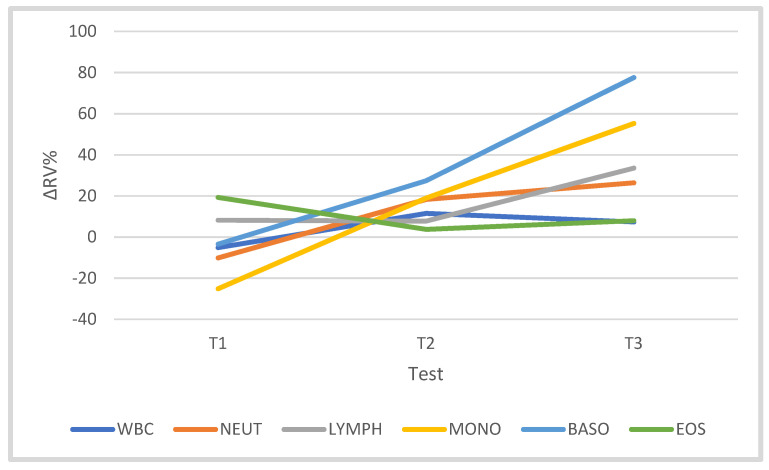
Changes (∆%) in leukocyte and leukocyte subset counts (%) in the blood of cyclists in tests T1, T2 and T3 in the training macrocycle, against baseline values (T0) of leukocyte subset.

**Figure 6 jcm-10-03299-f006:**
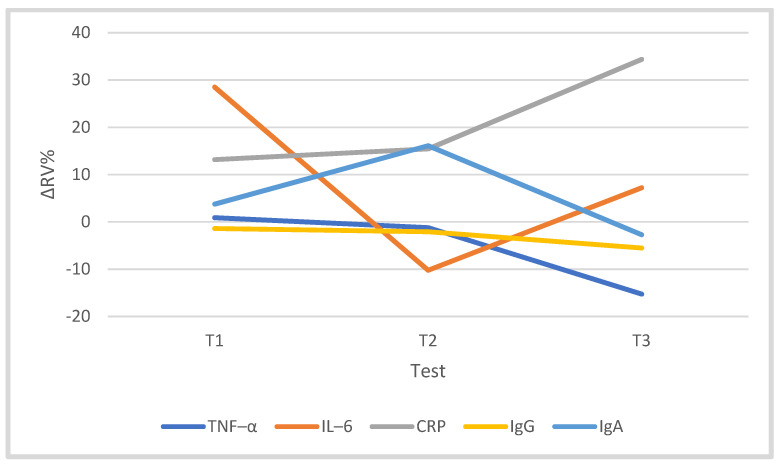
Changes (∆%) in immunological parameters in the blood of cyclists in the training macrocycle in tests T1, T2 and T3, in relation to the baseline study (T0). Abbreviations as in Table 3.

**Table 1 jcm-10-03299-t001:** The monthly average values of training loads (volume and intensity of effort, training strategies, aerobic and anaerobic training protocol). Baseline values (T0) and seasonal VO2max control tests (T1, T2, T3) of road cyclists.

Training Period/Tests	Month	Training Volume (h)	Training Volume (km)	Aerobic Zone	Aerobic–Anaerobic Zone	Anaerobic Zone
TRAN (T0)	November	49.04 ± 3.86	1183.00 ± 82.70	94	4	2
PREP (T1)	December	60.22 ± 6.29	1564.34 ± 116.26	90	6	4
	January	47.25 ± 5.05	1433.18 ± 78.75	70	25	5
	February	71.91 ± 7.90	2001.62 ± 244.02	65	27	8
	March	72.09 ± 8.09	2101.23 ± 185.06	62	23	15
COMP I (T2)	April	76.01 ± 7.55	2158.03 ± 103.51	60	20	20
	May	61.90 ± 3.08	2212.98 ± 113.00	55	25	20
COMP II (T3)	June	50.93 ± 3.52	2097.58 ± 117.48	50	25	25

Note: TRAN—after transition period, PREP—preparatory period, COMP I—start of competition period, COMP II—during competition period.

**Table 2 jcm-10-03299-t002:** Somatic–functional variables in the annual training cycle of cyclists. Registration during the TRAN period—end of the transition period (test—T0), PREP—preparation period (test—T1), COMP I—beginning of the competition period (test—T2), COMP II—during the competition period (test—T3).

**Variables/Period**	**TRAN; T0**		**PREP; T1**		**COMP I; T2**		**COMP II; T3**	
	X	S	X	S	X	S	X	S
BM (kg)	72.41	7.35	72.63	7.17	72.40	7.00	71.90	6.31
FAT (kg)	7.84	2.51	7.07	2.58	6.94	2.15	6.62	1.74
FAT (%)	10.73	2.78	9.58	3.04	9.50	2.53	9.17 *	2.13
BMI (kg/m^2^)	21.94	1.49	22.02	1.30	21.91	1.29	21.79	1.27
VO2max (mL/kg/min)	65.78	3.87	69.56	4.10	68.74	4.60	69.67	1.58
Biochemical variables in the annual training cycle of cyclists. Registration during the TRAN period—end of the transition period (test—T0), PREP—preparation period (test—T1), COMP I—beginning of the competition period (test—T2), COMP II—during the competition period (test—T3).								
**Variables/Period**	**TRAN; T0**		**PREP; T1**		**COMP I; T2**		**COMP II; T3**	
	X	S	X	S	X	S	X	S
MDA (µmol/L)	3.54	1.04	3.56	1.01	3.80	0.98	4.03	2.05
CK (U/L)	149.46	51.32	128.32	34.08	168.50	42.02	116.81 ^#^	29.27
LDH (U/L)	303.71	75.00	289.92	53.55	347.79	45.02	306.46	57.51
LA (mmol/L)	1.37	0.30	1.21	0.28	1.14	0.24	1.12	0.20

Abbreviations: BM—body weight, BMI—body mass index, FAT—fat content, VO2max—maximum oxygen uptake. Note: * statistically significant differences versus T0, *p* < 0.05. Abbreviations: MDA—malondialdehyde, CK—creatine kinase activity, LDH—lactate dehydrogenasesupp activity, LA—lactate level. Note: ^#^ T3 versus T2, *p* < 0.05.

**Table 3 jcm-10-03299-t003:** Leukocyte and leukocyte subset counts, immune system and the level of hormones in the blood of cyclists in the annual training cycle; exercise control tests T0—after the transition period, T1—the preparation period, T2—the beginning of the competition period, T3—during the competition period.

Variables/Period	TRAN; T0		PREP; T1		COMP I; T2		COMP II; T3	
	X	S	X	S	X	S	X	S
WBC (10^3^/µL)	5.52	1.18	5.12	0.73	6.11	1.79	5.80	0.74
Lymph (10^3^/µL)	2.14	0.62	2.21	0.51	2.24	0.57	2.83 *	0.74
Mono (10^3^/µL)	0.39	0.09	0.29 *	0.11	0.45	0.04	0.37	0.06
Neut (10^3^/µL)	2.80	0.85	2.40	0.36	3.21	1.40	3.00	0.62
Baso (10^3^/µL)	0.06	0.03	0.05	0.03	0.08	0.04	0.10 *	0.04
Eos (10^3^/µL)	0.15	0.06	0.17	0.05	0.15	0.06	0.14	0.07
TNF-α (pg/mL)	18.89	1.73	18.91	2.33	18.51	4.11	15.81	4.46
IL-6 (pg/mL)	0.55	0.34	0.56	0.18	0.46	0.24	0.62	0.53
CRP (mg/dL)	0.11	0.01	0.13	0.02	0.13	0.01	0.15	0.02
IgG (g/L)	9.25	1.89	9.13	1.93	9.07	1.97	8.67	1.71
IgA (g/L)	1.66	0.62	1.70	0.63	1.80	0.74	1.53	0.40
IGF-1 (ng/m)	218.38	61.63	214.84	74.66	188.20	68.92	239.52	36.62
GH (µg/L)	0.55	0.26	0.48	0.35	0.53	0.29	0.65	0.57
Cort (µg/dL)	19.62	5.54	19.46	3.55	21.04	4.69	17.49 ^#^	3.51
Test (ng/mL)	3.90	1.30	4.20	1.68	3.84	1.74	3.89	1.37
T/C	0.21	0.09	0.23	0.11	0.19	0.10	0.24	0.15
IR (µg/mL)	9.61	2.37	9.52	2.16	10.22	2.83	11.49	2.52

Abbreviations: WBC—leukocytes, Lymph—lymphocytes, Mono—monocytes, Neut—neutrophils, Baso—basophils, Eos—eosinophils, IGF-1—insulin-like growth factor-1, GH—growth hormone, Cort—cortisol, Test—testosterone, T/C—testosterone/cortisol ratio, IR—irisin, TNF-alfa—tumor necrosis factor alpha, IL-6—interleukin-6, CRP—C-reactive protein, IgA—immunoglobulin A, IgG—immunoglobulin G. Note: * statistically significant differences against T0, *p* < 0.05, ^#^ statistically significant differences against T2, *p* < 0.01.

**Table 4 jcm-10-03299-t004:** Pearson coefficient “r” between serum irisin (IR) (T0–T3) level and somatic–functional parameters in four stages of cyclists’ macrocycles (T0–T3).

**Parameter/IR**	**IR T0 vs.**	**IR T1 vs.**	**IR T2 vs.**	**IR T3 vs.**
BM *(T0–T3)	0.101; *p* = 0.796	−0.199; *p* = 0.609	0.174; *p* = 0.65	0.612; *p* = 0.080
BMI (T0–T3)	0.267; *p* = 0.488	−0.211; *p* = 0.586	−0.165; *p* = 0.671	0.588; *p* = 0.096
FAT kg (T0–T3)	0.222; *p* = 0.566	0.003; *p* = 0.994	0.228; *p* = 0.556	0.495; *p* = 0.176
FAT% (T0–T3)	0.264; *p* = 0.492	0.058; *p* = 0.882	0.234; *p* = 0.545	0.331; *p* = 0.385
VO2max (T0–T3)	−0.441; *p* = 0.235	−0.482; *p* = 0.189	−0.101; *p* = 0.796	−0.216; *p* = 0.578
CK (T0–T3)	−0.319; *p* = 0.401	−0.519; *p* = 0.152	0.353; *p* = 0.351	0.523; *p* = 0.148
LDH (TO–T3)	−0.356; *p* = 0.342	0.448; *p* = 0.227	−0.158; *p* = 0.685	0.372; *p* = 0.324
MDA (T0–T3)	−0.035; *p* = 0.930	−0.323; *p* = 0.397	0.333; *p* = 0.381	0.168; *p* = 0.666
LA (T0–T3)	−0.163; *p* = 0.675	0.245; *p* = 0.525	−0.224; *p* = 0.562	0.156; *p* = 0.688
Pearson coefficient “r” between serum irisin (IR) level and WBC components, selected hormone concentrations and immunological factors in four stages of cyclists’ macrocycles (T0–T3).				
**Parameter/IR**	**IR T0 vs.**	**IR T1 vs.**	**IR T2 vs.**	**IR T3 vs.**
WBC (T0–T3)	−0.433; *p* = 0.244	−0.418; *p* = 0.263	−0.413; *p* = 0.269	−0.455; *p* = 0.218
Lymph (T0–T3)	−0.339; *p* = 0.372	−0.620; *p* = 0.075	−0.411; *p* = 0.272	−0.269; *p* = 0.483
Mono (T0–T3)	−0.159; *p* = 0.683	−0.326; *p* = 0.391	−0.111; *p* = 0.777	−0.495; *p* = 0.176
Neut (T0–T3)	−0.286; *p* = 0.455	0.105; *p* = 0.787	−0.362; *p* = 0.38	−0.173; *p* = 0.656
Baso (T0–T3)	0.011; *p* = 0.977	−0.223; *p* = 0.564	−0.646; *p* = 0.060	−0.344; *p* = 0.365
Eos (T0–T3)	−0.649; *p* = 0.058	0.085; *p* = 0.828	0.223; *p* = 0.564	−0.177; *p* = 0.649
Test (T0–T3)	0.309; *p* = 0.418	0.178; *p* = 0.646	−0.049; *p* = 0.901	−0.238; *p* = 0.537
Cort (T0–T3)	0.758; *p* = 0.018 *	0.304; *p* = 0.427	0.545; *p* = 0.129	0.544; *p* = 0.130
T/C (T0–T3)	−0.220; *p* = 0.569	0.043; *p* = 0.913	−0.278; *p* = 0.469	−0.345; *p* = 0.363
GH (T0–T3)	−0.230; *p* = 0.551	−0.759; *p* = 0.018 *	−0.567; *p* = 0.111	−0.527; *p* = 0.145
IGF1 (T0–T3)	−0.417; *p* = 0.265	−0.232; *p* = 0.548	−0.236; *p* = 0.54	−0.156; *p* = 0.689
TNFα (T0–T3)	−0.303; *p* = 0.428	0.073; *p* = 0.851	−0.559; *p* = 0.118	−0.466; *p* = 0.206
IL-6 (T0–T3)	−0.369; *p* = 0.328	0.289; *p* = 0.450	0.476; *p* = 0.195	−0.053; *p* = 0.892
CRP (T0–T3)	−0.152; *p* = 0.423	0.037; *p* = 0.924	0.00; *p* = 1.00	0.119; *p* = 0.761
IgA (T0–T3)	−0.058; *p* = 0.882	0.203; *p* = 0.601	0.092; *p* = 0.813	0.551; *p* = 0.124
IgG (T0–T3)	0.325; *p* = 0.394	0.115; *p* = 0.768	0.188; *p* = 0.628	0.013; *p* = 0.974
WBC (T0–T3)	−0.433; *p* = 0.244	−0.418; *p* = 0.263	−0.413; *p* = 0.269	−0.455; *p* = 0.218
Lymph (T0–T3)	−0.339; *p* = 0.372	−0.620; *p* = 0.075	−0.411; *p* = 0.272	−0.269; *p* = 0.483

Abbreviations as in Table 2 and Table 3. Note: * T0–T3—check steps in the annual cycle. η^2^p size effect: small, 0.01–0.06; medium, 0.06, strong 0.14.

## Data Availability

Data are available on request due to privacy and ethical restrictions.

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
