# Peer review of "Effect of Seasonal Variation during Annual Cyclist Training on Somatic Function, White Blood Cells Composition, Immunological System, Selected Hormones and Their Interaction with Irisin"

_jcm, 2021, doi:10.3390/jcm10153299_

Round 1
Reviewer 1 Report
Thank you for inviting me to review this manuscript. At the beginning, the authors of the work presented the abstract in a too general manner,and not presenting the research issues and methodology in a very interesting way. Abstract correction is needed - please readthe guidelines for authors.Then the study presents wide observation results of only 9 men, which requires marking this work as preliminary research or pilot study. Please include this information in the title. Subsequently, there is no information on the selection of the research group (method of recruitment, research criteria). I am asking for such information. There is also no information about No of the bioethical commission approvement for this research. Please let me know why the authors only studied men?
There is no information on the diet of the studied group in the methodology and discussion. This is very important in the context of such extensive biochemical research. Please complete this aspect of the study group assessment. Overall, the study was very interesting, although performed on a small group of participants. Therefore, all summaries require only its initial, pilot significance to be marked.
Reviewer 2 Report
The paper “Effect of seasonal variation during an annual cyclists training…” presents an interesting and valuable study describing somatic, hormonal and immunological changes occurring in cyclists from the perspective of the assessment of the training macrocycle. The study has been carefully prepared and I have no objections to the methodology used; the study limitations were indicated by the authors. Below some minor comments:
- The Authors used the template for "diagnostics" instead of the journal JCM.
- Line 51: I think the sentence "This hormone reduces the level of TNF-α and probably other inflammatory factors [22]." should be redrafted as it is commonly known in literature that irisin has both anti-inflammatory and antioxidant effects. Therefore, the term "probably" is not appropriate here. Also, the reference [22] refers to the study where the anti-inflammatory effect of irisin was not investigated.
- Abbreviations should be explained in the first place of their occurrence in the paper ̶— separately in the main text and under figures / graphs.
- In my opinion, spatial charts are not an adequate form of presentation of scientific data, those charts reduce the readability of the data. Please edit Figure 1. Whether there were statistically significant differences between the groups should be noted.
- The charts must have clearly signed Y axes with units indicated (see Fig. 2 [%]). There is no need to redundantly repeat "%" for each number. In addition, the principle of self-explanation of charts and tables forces us to fully and completely describe the information under the chart. Unfortunately, on the chart I do not see if the groups are statistically equal to each other.
- Figures 3-5 are valuable graphs, but in their current form they are difficult to read, maybe it is worth applying colors to help their interpretation.
Round 2
Reviewer 1 Report
I accept it as it is.